# Exploring the Influence of Ecological Niches and Hologenome Dynamics on the Growth of *Encephalartos villosus* in Scarp Forests

Nqobile Motsomane [1], Terence N. Suinyuy [2], María A. Pérez-Fernández [3] and Anathi Magadlela [1,4,*]

1 School of Life Sciences, College of Agriculture, Engineering and Science, University of KwaZulu-Natal (Westville Campus), Private Bag X54001, Durban 4000, South Africa; 217054036@stu.ukzn.ac.za
2 School of Biology and Environmental Sciences, University of Mpumalanga (Mbombela Campus), Private Bag X11283, Mbombela 1200, South Africa; terence.suinyuy@ump.ac.za
3 Department of Physical, Chemical and Natural Systems, Universidad Pablo de Olavide, Carretera de Utrera Km 1, 41013 Seville, Spain; maperfer@upo.es
4 School of Natural and Applied Sciences, Sol Plaatje University, Private Bag X5008, Kimberly 8300, South Africa
* Correspondence: anathimagadlela@icloud.com

**Abstract:** Information on how bacteria in plants and soil, along with extracellular enzymes, affect nutrient cycling in *Encephalartos villosus* growing in phosphorus deficient and acidic scarp forests is lacking. Bacteria in coralloid roots, rhizosphere, and non-rhizosphere soils were isolated to determine the potential role of soil bacterial communities and their associated enzyme activities in nutrient contributions in rhizosphere and non-rhizosphere soils. The role of soil characteristics and associated bacteria on *E. villosus* nutrition and nitrogen source reliance was investigated. *Encephalartos villosus* leaves, coralloid roots, rhizosphere, and non-rhizosphere soils were collected at two scarp forests. Leaf nutrition, nitrogen source reliance, soil nutrition, and extracellular enzyme activities were assayed. A phylogenetic approach was used to determine the evolutionary relationship between identified bacterial nucleotide sequences. The clustering pattern of isolated bacterial strains was primarily dictated by the ecological niches from which they originated (rhizosphere soil, non-rhizosphere soil, and coralloid roots), thus indicating that host-microbe interactions may be a key driver of this pattern, in line with the hologenome theory. There were insignificant differences in the phosphorus and nitrogen cycling enzyme activities in *E. villosus* rhizosphere and non-rhizosphere soils in both localities. Significantly positive correlations were recorded between nitrogen and phosphorus cycling enzymes and phosphorus and nitrogen concentrations in rhizosphere and non-rhizosphere soils. Additionally, more than 70% of the leaf nitrogen was derived from the atmosphere. This study challenged the conventional expectation that environmental filters alone dictate microbial community composition in similar habitats and revealed that host-microbe interactions, as proposed by the hologenome theory, are significant drivers of microbial community structuring. The isolated bacteria and their plant growth promoting traits play a role in *E. villosus* nutrition and nitrogen source reliance and secrete nutrient cycling enzymes that promote nutrient availability in rhizosphere and non-rhizosphere soils.

**Keywords:** cycad-microbe symbiosis; phosphorus deficient soils; biological nitrogen fixation





## 1. Introduction

Cycads are ancient gymnosperms that have persisted through many extinction events, survived competition with fast-growing angiosperms, and persisted through environmental stresses such as drought and nutrient deficiencies [1,2]. The Zamiaceae is the most diverse cycad family in South Africa [3]. The *Encephalartos* genus belongs to the Zamiaceae family and has 37 species that are indigenous to South Africa, making South Africa an important centre for cycad diversity [4,5]. *Encephalartos villosus* is a forest understory cycad and is

among the most diverse and widely distributed species in the *Encephalartos* genus [6,7]. According to [8], cycads are distributed in grasslands and scarp forests that are characterised as nutrient deficient [9]. The growth and development of cycads in these ecosystems may be attributed to their ability to form symbiotic associations with plant growth promoting bacteria [8] and their role in enhancing soil nutrients [10]. According to [11], cycads offer essential ecosystem services such as nutrient cycling, including carbon sequestration, soil conservation, wildlife habitat provision, and medicinal resource utilization. Cycads absorb carbon dioxide from the atmosphere through photosynthesis, helping to mitigate climate change by reducing atmospheric greenhouse gas levels [12]. Their extensive root systems stabilize soil and prevent erosion, improving water quality and reducing the risk of landslides [13]. Additionally, cycads serve as a food source for numerous animal species, supporting local biodiversity and contributing to the overall health of ecosystems [14]. Finally, some cycad species have traditional medicinal uses, highlighting their potential as a renewable resource for human wellbeing [15].

Ref. [10] reported higher nitrogen (N), carbon (C), and phosphorus (P) concentrations in *Cycas micronesica* and *Zamia integrifolia* rhizosphere soils compared to non-rhizosphere soil, highlighting the soil nutrient contributions by cycads. Ref. [11] reported that soil bacterial communities and their associated enzyme activities contribute to soil nutrient inputs, enabling *Encephalartos natalensis* to thrive in nutrient deficient and acidic ecosystems. Thus, we can deduce that the role of cycads on soil nutrient inputs may be due to cycad-microbe symbiosis, soil bacterial communities, and associated enzyme activities [11]. The bioavailability of soil nutrients depends on soil bacteria that release extracellular enzymes to mineralise and cycle nutrients, making them available for plant assimilation [16,17]. Soil bacteria exude enzymes, such as phosphatase, N-acetylglucosaminidase, and β-glucosidase, that play a role in the mineralisation and cycling of P, N and C [18,19]. Studies such as [10] report that cycads improve soil N, C, and P content, but the role of soil bacterial communities and their associated enzyme activities is not accounted for.

According to [11], soil bacterial communities and their associated enzymes enhance nutrient bioavailability in rhizosphere and non-rhizosphere soils. Therefore, soil bacterial communities and their associated enzymes may influence the growth of *E. villosus* growing in the scarp forests of the Eastern Cape, South Africa. The growth of cycad species in nutrient deficient ecosystems is believed to be due to cycad-microbe symbiosis. Cycads are reportedly associated with plant growth-promoting (PGP) bacteria belonging to the *Rhizobium*, *Lysinibacillus*, *Bradyrhizobium*, and *Burkholderia* genera [11,20]. These microbes assist cycad growth by providing essential elements such as nitrogen (N) and phosphorus (P) [21], enabling cycads species to grow in nutrient deficient ecosystems such as scarp forests. Ref. [22] reported that symbiotic associations involving cycads and PGP bacteria produce insignificant variations in the N and P concentrations of cycad leaves, highlighting the role that cycad-microbe symbiosis plays on plant nutrition. Though a comprehensive review of the elemental composition of cycad leaves has been compiled by [23], the role of PGP bacteria on *E. villosus* nutrition is yet to be studied. Also, many studies reported on the N-fixing bacteria associated with cycad species [20,24,25]; however, the influence of the bacterial isolates on plant nutrition and N source reliance is poorly understood. Moreover, studies that report on the composition of the $^{15}$N isotope in cycad foliage, such as [26], do not report on the N reliance and the percentage of N derived from the atmosphere in cycads.

This study isolated and identified bacteria in *E. villosus* coralloid roots, rhizosphere, and non-rhizosphere soils and determined the role of soil characteristics and associated bacteria on *E. villosus* nutrition and N source reliance. The study also investigated the potential role of soil bacterial communities and their associated enzyme activities in nutrient contributions in *E. villosus* rhizosphere and non-rhizosphere soils. The objectives of this study included (1) determining the soil characteristics (pH, exchange acidity, total cation, and nutrient concentrations) in *E. villosus* rhizosphere and non-rhizosphere soils, (2) verifying the association of N-fixing, P-solubilising, and N-cycling bacteria found in

*E. villosus* coralloid roots, rhizosphere and non-rhizosphere soils on the growth of *E. villosus* in the two scarp forests, (3) assaying the soil nutrient (P and N) cycling enzyme activities in *E. villosus* rhizosphere and non-rhizosphere soils, and (4) using $^{15}$N isotope to determine the N reliance of *E. villosus* growing in nutrient deficient and acidic scarp forest ecosystem soils. We hypothesized that the bacterial composition in coralloid roots, rhizosphere, and non-rhizosphere soils of *E. villosus* and associated extracellular enzyme activities would indicate the species' contribution to the bioavailability of soil nutrition and contribute to plant growth, nutrition, and reliance on N derived from the atmosphere. Conducting a study on the role of cycad-associated microbes on *E. villosus* growth and nutrition is essential because *E. villosus* has a life history similar to cycads that cannot be studied due to their conservation status. Therefore, studying the role of microbes on *E. villosus* growth will not only shed light on how microbial interactions in *E. villosus* contribute to soil nutrient inputs, growth, and plant nutrition, but will also give insights into other cycad species with similar life history.

## 2. Materials and Methods

### 2.1. Study Sites and Species

Sixteen randomly selected adult *E. villosus* plants of the same age, growing in two scarp forests, were sampled. The scarp forests are Oceanview Farm in East London and Rhebu village in Port St Johns (eight plants were sampled from each locality), Eastern Cape, South Africa (coordinates not included due to cycads conservation concerns). These two scarp forests, like others, occur in nutrient-poor, leached, and shallow soils [9]. The forest floor is covered with decaying leaflitter, with *E. villosus* as the dominant understory plant and tall trees, such as *Ptaeroxylon obliquum* (F.Muell.) Benth. and Hook.f. ex Harv., *Hyperacanthus amoenus* L., *Protorhus longifolia* (Lam.) Small., and *Vepris undulata* (Aiton) Radlk., adorning the forests [9]. In addition to muthi (traditional healer) harvesting and collection of biofuels, the scarp forests in Oceanview and Rhebu village are also disturbed by cattle that graze within the forests. In the process, the cattle deposit dung and urine, enriching the forest soils. According to [27], the chemical composition of rhizosphere soils in the root and litterfall zone may be altered by cycads. Thus, rhizosphere soils were collected directly beneath the selected plants and along the dripline of the leaf canopy. The soils were collected at 0–20 cm depth at the four cardinal points of each plant (North, West, South, and East). Similarly, non-rhizosphere soils (similar depths) were collected from non-target sites defined by a radius of five meters from the base of each target plant. The direction of the non-rhizosphere samples was randomly selected using cardinal points of North, East, South, and West of the target plant. From each *E. villosus* target plant, soil samples (rhizosphere and non-rhizosphere) collected at different depths and cardinal points were bulked to form composite samples for soil characteristic analysis (nutrient concentrations, total cation, exchange acidity, and pH).

### 2.2. Soil Characteristics

Ref. [28] reported that soil nutrient concentrations directly affect cycad leaf concentrations. Therefore, thirty-two 50 g compound soil samples collected from each sampled *E. villosus* plant were sent to the KwaZulu Natal Department of Agriculture and Rural Development Analytical services for soil nutrient concentrations, total cation concentrations, exchange acidity, and pH analysis. The soil characteristic analysis was performed per protocols explained in [29]. Ambic-2 solution containing 0.25 M NH$_4$CO$_3$, 0.01 M Na$_2$EDTA, 0.01 M NH$_4$F, and 0.05 g/L superfloc (N100) was adjusted to pH 8 using concentrated ammonia solution and used to extract phosphorus, potassium, zinc, copper, and manganese. The extracts were filtered using Whatman no. 1, and a 2 mL filtrate aliquot was used to determine the phosphorus concentration using a modified protocol of [30] molybdenum blue procedure. The potassium concentration was determined by diluting 5 mL aliquot of the filtrate with 20 mL de-ionised water using atomic absorption, and the remaining undiluted filtrate was used to determine the zinc, copper, and manganese

concentration using atomic absorption. The magnesium and calcium concentrations were determined by stirring sample cups containing 25 mL of soil sample and 25 mL of 1 M KCl solution in a multiple stirrer (400 rpm) for 10 min. The stirred mixture was filtered with Whatman no.1 paper. Five mililitres of the filtrate was diluted with 20 mL 0.0356M SrCl$_2$, and calcium and manganese concentrations were determined using atomic absorption spectrophotometer. The extractable acidity was determined by diluting 10 mL of the filtrate with 10 mL of de-ionised water containing 2-4 drops of phenolphthalein and titrated with 0.005 M NaOH. Soil nitrogen concentration was measured using the Automated Dumas dry combustion method with a LECO CNS 2000 (Leco Corporation, Michigan USA). Soil samples were weighed in a ceramic crucible, and 0.5 g vanadium pentaoxide was used as a combustion catalyst. The crucible was placed in a horizontal furnace and burned in a stream of oxygen at 1350 °C, and soil N was measured as N$_2$ in a thermal conductivity cell. Soil pH was determined by mixing 10 mL of soil sample and 25 mL of 1 M KCl in sample cups and stirring in a multiple stirrer at 400 rpm for 5 min. The suspension was left to rest for 30 min, and the pH was measured using a gel-filled combination glass electrode while stirring.

### 2.3. Coralloid Root Surface Sterilisation

Coralloid roots were harvested from eight randomly selected mature *E. villosus* individuals in Rhebu and Oceanview, respectively. The coralloid roots were surface sterilized with 70% (*v/v*) ethanol for 30 s and soaked in 3.5% (*v/v*) sodium hypochlorite solution for 3 min. Thereafter, they were rinsed ten times with sterile distilled water and immersed in 100 μL of 15% glycerol in sterile Eppendorf tubes.

### 2.4. Bacterial Extraction and Identification from Coralloid Roots and Soils

Soil samples collected from the rhizosphere and non-rhizosphere soils in Rhebu and Oceanview were subjected to serial dilutions, and the immersed coralloid roots were crushed using sterile pipette tips for bacterial extraction [30]. Ten μL of the root suspension and 50 μL of each serial dilution were cultured in sterile Petri plates containing selective media (Pikovskaya's plate containing tricalcium phosphate (TCP) for P-solubilizing bacteria, Simmons citrate agar for N-cycling bacteria, and Jensen's media agar for N-fixing bacteria) to ensure that only bacteria with P-solubilizing, N-cycling, and N-fixing traits grew on the plates [31]. Each selective media was replicated three times and incubated at 30 °C for five days. Pure bacterial colonies were obtained by repeated streaking/subculturing. A small portion of the pure bacterial colonies was amplified through polymerase chain reaction (PCR) using the 16S ribosomal RNA gene primers: 63F (5′ CAGGCCTAACACATGCAAGTC 3′ (21 bases) and 1387R (5′ GGGCGGTGTGTACAAGGC 3′ (18 bases)) [32] from Inqaba Biotechnical Industries (Pty) Ltd. (South Africa). The PCR amplification was performed using an EmaraldAmp GT Master Mix with the following conditions: Initial denaturation at 94 °C for 5 min, followed by 30 cycles of denaturation at 94 °C for 30 s, annealing at 55 °C for 30 s and extension at 72 °C for 2 min, with additional extension at 72 °C for 10 min. The PCR products were separated by electrophoresis on 1% (*w/v*) agarose gel and visualized under UV light to determine the correct product size amplification. The amplicons were sent for sequencing at Inqaba Biotechnical Industries (Pty) Ltd., Pretoria South Africa. The DNA sequences were edited and compared to the nucleotide sequences of known bacteria in the GenBank database of the National Centre for Biotechnology Information (NCBI) by using Basic Local Aligned Search Tool (BLAST) (https://www.ncbi.nlm.nih.gov, assessed on 16 June 2023).

Using the sequences of the identified bacteria, a phylogenetic approach was used to determine the evolutionary relationship between bacterial nucleotide sequences. The nucleotide alignment was conducted using the MUSCLE tool in MEGA 11 and checked manually before constructing the phylogenetic tree using the neighbour-joining likelihood tree approach. A bootstrap resampling was performed with 1000 replicates in accordance with the procedure in [33].

### 2.5. Extracellular Enzyme Activities

N-acetylglucosaminidase, acid phosphatase and alkaline phosphatase activities were assayed using the fluorescence-based method described by [34] and expressed as nmol h$^{-1}$ g$^{-1}$. Briefly, samples (10 g soil/100 mL autoclaved dH$_2$O) were homogenised at medium speed in a shaker for two hours and stored overnight at 4 °C. The supernatants were transferred into black 96-well microplates before adding their respective substrates. The sample run consisted of 200 µL soil aliquot and 50 µL substrate, alongside reference standards (200 µL bicarbonate buffer + 50 µL standard), quench standard (200 µL soil aliquot + 50 µL standard), sample control (200 µL soil aliquot + 50 µL buffer), negative controls (200 µL buffer + 50 µL substrate), and blanks (250 µL buffer). The 96-well plate was incubated at 25 °C for 2 h. The reaction was stopped afterward by adding 5 µL of 0.5 M NaOH to each well. The fluorescence was measured at 450 nm on a Glomax Multi Plus microplate reader. The buffer and standard were adjusted to pH 5 before determining acid phosphatase activity. Nitrate reductase activity assays were performed using a modified protocol described by [35]. A volumetric flask wrapped in foil was filled with 1 mL of 25 mM KNO$_3$, 4 mL of 0.9 mM 2,4-dinitrophenol, and 5 mL of milliQ dH$_2$O. Thereafter, 5 g of soil was added to the solution, and the flask was sealed with foil, shaken, and incubated in an oven at 30 °C for 24 h. After incubation, 10 mL of 4 M KCl was added to the soil mixture, succinctly mixed, and filtered using filter paper (Whatman number 1). The enzymatic reaction was initiated by adding 2 mL of the filtrate to 1.2 mL of 0.19 M ammonium chloride buffer (pH 8.5) and 0.8 mL of a colour reagent consisting of 1% sulfanilamide, 1N HCl, and 0.02% N-(1-naphthyl) ethylenediamine dihydrochloride (NEDD). The solution was incubated at 30 °C for 30 min. The absorbance was measured at 520 nm using an 1800 UV spectrophotometer. The nitrite (NO$_2^-$) liberated into the medium was extrapolated from a prepared standard curve with KNO$_3$. The nitrate reductase activity was expressed as 0.1 µmol h$^{-1}$ g$^{-1}$.

### 2.6. Leaf Nutrient Composition

Ref. [36] reported that leaf nutrient compositions differ with age. Therefore, young leaves were sampled. Leaf N and P concentrations were considered to determine the role of N-fixing, N-cycling, and P solubilising bacteria on leaf nutrition. The number of leaves and leaf length (cm) from selected plants were measured. Leaves from *E. villosus* plants of the same size in Rhebu and Oceanview were collected on the leaf's basal, midpoint, and apical locations per [22]. The sampled leaves were oven dried at 80 °C till constant weight, ground into a fine powder with a tissuelyser, and sent to the Central Analytical Facilities at the University of Stellenbosch (South Africa) for P and N analysis through Inductively Coupled Mass Spectrometry (ICP-MS) and N isotope analysis at the Archeometry Department at the University of Cape Town (South Africa).

### 2.7. Percentage N Derived from the Atmosphere (%NDFA)

The University of Cape Town Archeometry Department conducted the N isotope analysis. The isotopic ratio of N was calculated as δ = 1000 (Rsample/Rstandard), where R is the molar ratio of the heavier to the lighter isotope of the samples and standards. Between 2.10 and 2.20 mg of each milled sample were weighed into 8 mm × 5 mm tin capsules (Elemental Micro-analysis, Devon, UK) on a Sartorius microbalance (Goettingen, Germany). The samples were then combusted in a Fisons NA 1500 (Series 2) CHN analyser (Fisons Instruments SpA, Milan, Italy). The nitrogen isotope values for the N gas released were determined on a Finnigan Matt 252 mass spectrometer (Finnigan MAT GmbH, Bremen, Germany), which was connected to a CHN analyser by a Finnigan MAT Conflo control unit. Five standards were used to correct the samples for machine drift, namely, two in-house standards (Merck Gel and Nasturtium) and the IAEA (International Atomic Energy Agency) standard (NH$_4$)$_2$SO$_4$.

%NDFA = 100 ((δ$^{15}$N reference plant − δ$^{15}$N cycad)/(δ$^{15}$ N reference plant − β))

where NDFA is the N derived from the atmosphere and the B value represents the δ15N natural abundance of the N derived from biological $N_2$ fixation. The %NDFA was represented as a percentage [37].

### 2.8. Statistical Analysis

R studio version 3.6.2 was used to test for differences in the micronutrients, pH, exchange acidity, soil enzyme activity, and total cation in soil samples collected in *E. villosus* rhizosphere and non-rhizosphere soils using independent samples *T*-test. The assumptions for normality and homogeneity of the variances were tested using the one-sample Kolmogorov–Smirnov normality and Levene's test, respectively ($p > 0.05$). The Wilcoxon test, a non-parametric alternative, was used in cases where the assumptions were not met. A simple linear correlation analysis was performed to determine if there is a relationship between soil enzyme activities and soil nutrition. A probability of $p \leq 0.05$ was considered significant. A simple linear correlation analysis was conducted to determine if there is a relationship between soil enzyme activities and soil nutrition. A probability of $p \leq 0.05$ was considered significant.

## 3. Results

### 3.1. Soil Characteristics

The magnesium (Mg) and zinc (Zn) concentrations in rhizosphere soils were significantly higher than in the non-rhizosphere soils in Oceanview (Table 1). There were no significant differences in the macronutrient (nitrogen (N), phosphorus (P), and potassium (K)) concentrations of rhizosphere and non-rhizosphere soils in both localities (Table 1). The soil samples collected from rhizosphere and non-rhizosphere soils in Rhebu and Oceanview were relatively acidic (Table 1). In Rhebu and Oceanview, rhizosphere soils had a higher pH of 4.28 and 5.43 compared to pH of 4.14 and 4.97 for the non-rhizosphere soils, respectively (Table 1).

**Table 1.** Soil nutrient concentration and relative acidity of *Encephalartos villosus* rhizosphere and non-rhizosphere soils in Oceanview and Rhebu, Eastern Cape, South Africa.

| Study Site | Oceanview | | Rhebu | |
|---|---|---|---|---|
| Parameter | Rhizosphere | Non-Rhizosphere Soils | Rhizosphere | Non-Rhizosphere Soils |
| P (mg/kg) | 6.1 ± 2.9 [a] | 3.86 ± 1.9 [a] | 22.95 ± 5.9 [a] | 13.23 ± 3.7 [a] |
| K (cmolc/kg) | 139.5 ± 33.6 [a] | 81.9 ± 12.5 [a] | 330.0 ± 100.9 [a] | 179.1 ± 54.9 [a] |
| N (mg/kg) | 3093.7 ± 741.9 [a] | 2638.0 ± 438.2 [a] | 5189.3 ± 297.9 [a] | 3473.3 ± 1396.3 [a] |
| Ca (cmolc/kg) | 1555.3 ± 274.8 [a] | 1379.2 ± 153.5 [a] | 1534.9 ± 974.2 [a] | 1229.4 ± 1087.2 [a] |
| Mg (cmolc/kg) | 454.7 ± 46.1 [a] | 382.12 ± 35.3 [b] | 838.5 ± 549.3 [a] | 697.1 ± 548.2 [a] |
| Mn (mg/kg) | 55.3 ± 21.1 [a] | 44.8 ± 15.9 [a] | 89.7 ± 58.0 [a] | 48.1 ± 20.2 [a] |
| Zn (mg/kg) | 2.2 ± 0.1 [c] | 1.3 ± 0.4 [d] | 4.1 ± 1.6 [a] | 2.8 ± 1.2 [a] |
| Cu (mg/kg) | 1.1 ± 0.2 [a] | 0.6 ± 0.1 [a] | 0.9 ± 0.1 [a] | 0.6 ± 0.2 [a] |
| Exchange acidity (cmol/L) | 0.05 ± 0.01 [a] | 0.05 ± 0.01 [a] | 1.46 ± 1.42 [a] | 1.37 ± 1.19 [a] |
| Total cations (cmol/L) | 11.53 ± 1.65 [a] | 11.65 ± 1.62 [a] | 15.26 ± 8.13 [a] | 15.10 ± 9.48 [a] |
| pH | 5.43 ± 0.41 [e] | 4.97 ± 0.57 [f] | 4.28 ± 0.61 [a] | 4.14 ± 0.58 [a] |

In each row, different letters denote significant differences in the nutrient concentrations and relative acidity of soil samples within study sites (independent samples *t*-test, $p \leq 0.05$, means ± SE, *n* = 32).

### 3.2. Bacterial Identification

Three phylogenetic trees were generated based on bacterial isolates collected from Oceanview (Figure 1), Rhebu (Figure 2), and a combination of both sites (Figure 3). Bacteria from rhizosphere soils from both Oceanview and Rhebu were in 20 different genera, whereas those isolated from non-rhizosphere soils were in 12 genera, and only 7 genera were identified from coralloid roots (Figures 1–3). Notably, bacterial clusters were found to be associated with specific sampling locations, including rhizosphere soil, non-rhizosphere soil, and coralloid roots. While a few exceptions were observed, particularly in the case of the Oceanview sample (Figure 1), the clustering pattern was consistently evident across all three datasets, indicating that the origin of the bacterial isolates significantly influenced their phylogenetic profiles.

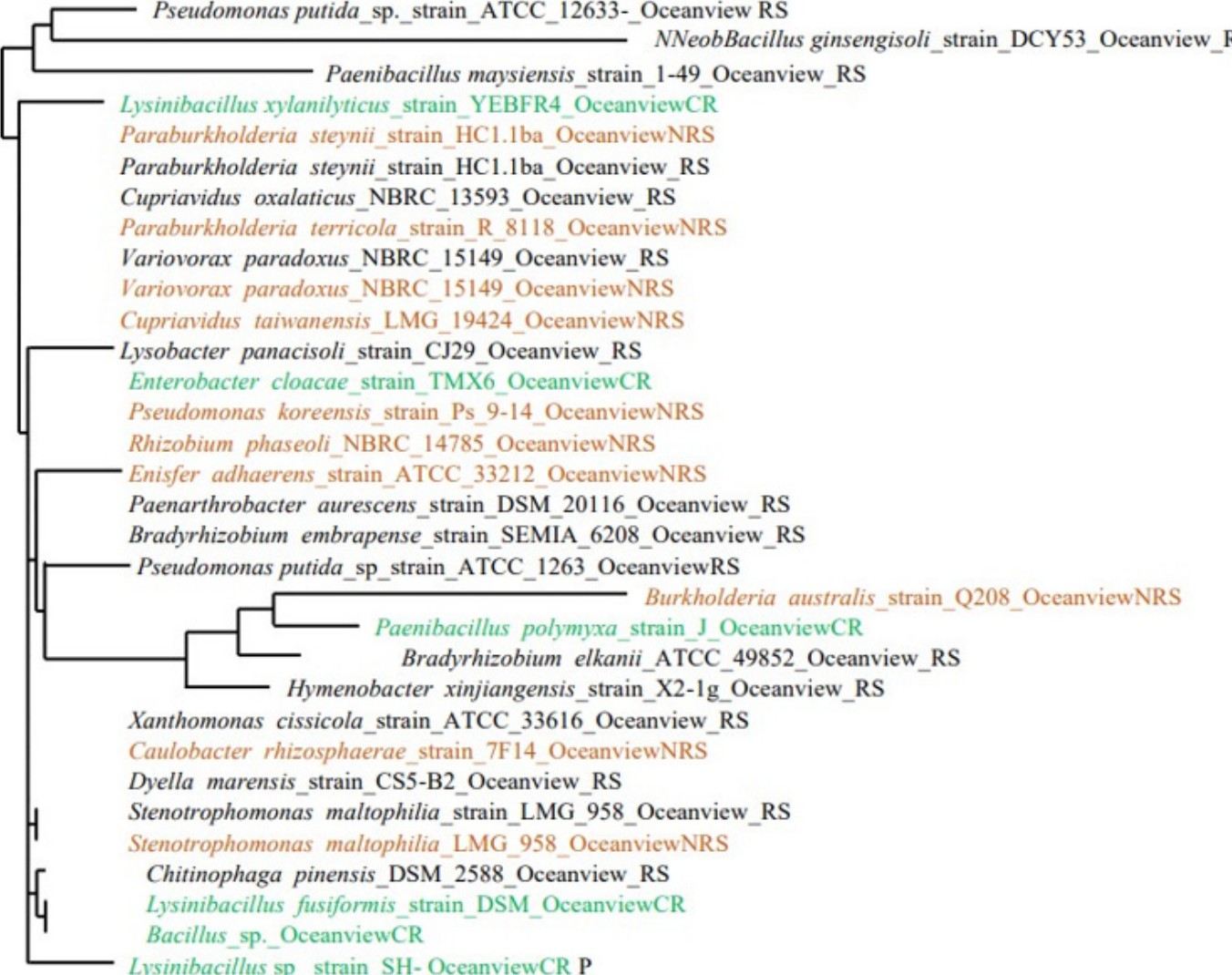

**Figure 1.** Phylogenetic placement of the 16S rRNA gene from 32 isolates from rhizosphere soil (black), non-rhizosphere soil (brown) and coralloid roots (green) of *Encephalartos villosus* grown in Oceanview in the Eastern Cape Province, South Africa.

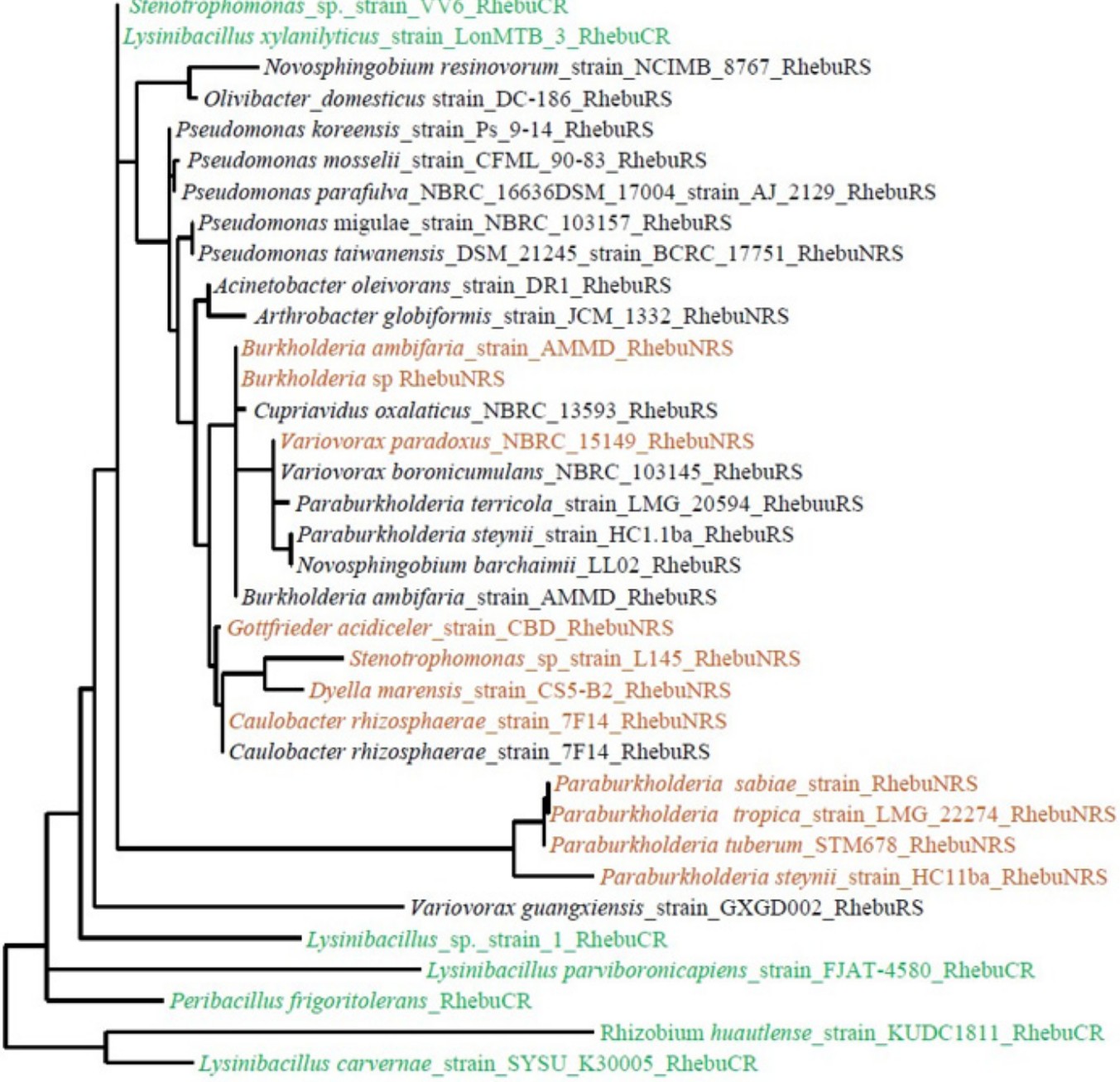

**Figure 2.** Phylogenetic placement of the 16S rRNA gene from 36 isolates from rhizosphere soil (black), non-rhizosphere soil (brown) and coralloid roots (green) of *Encephalartos villosus* grown in Rhebu in the Eastern Cape Province, South Africa.

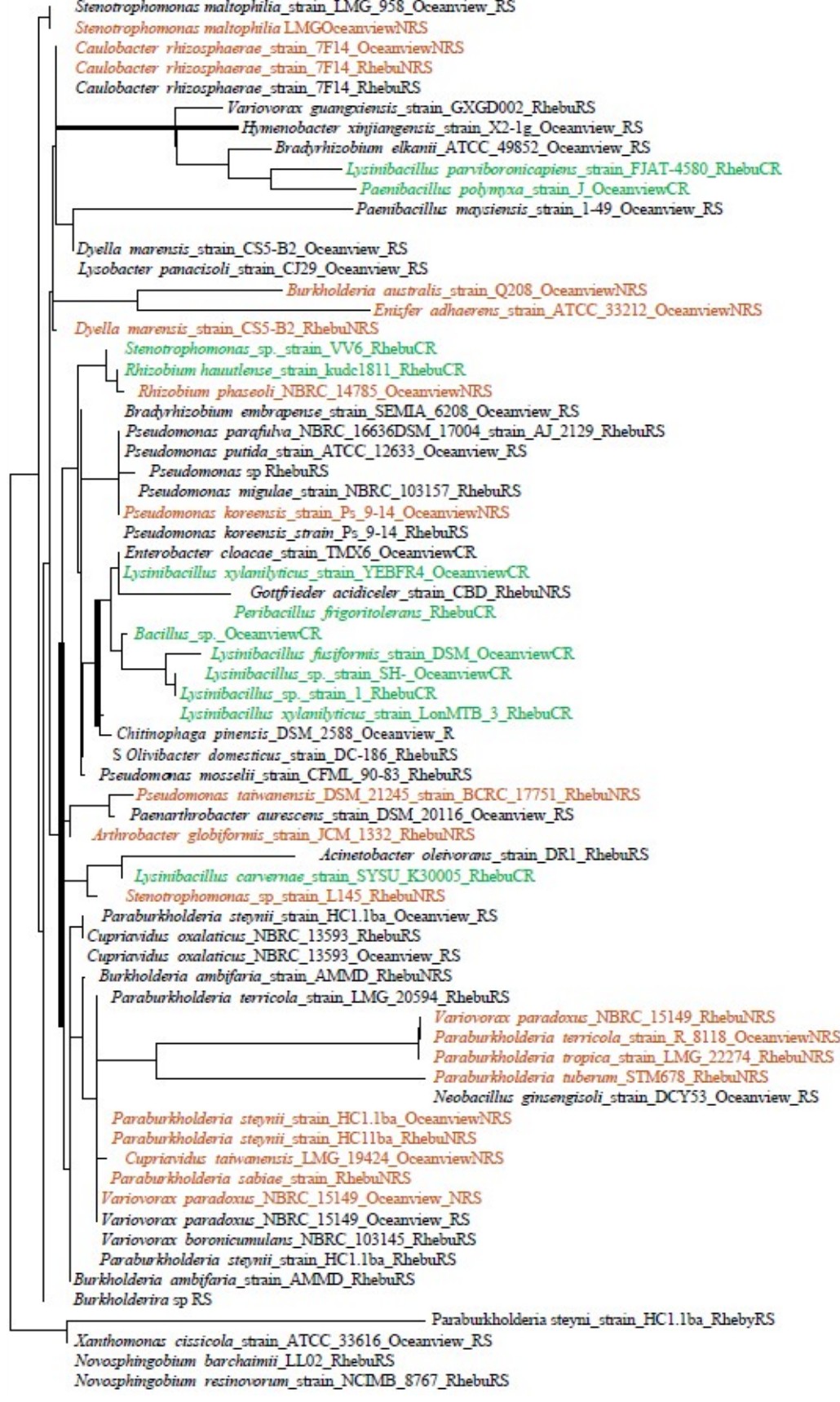

**Figure 3.** Phylogenetic placement of the 16S rRNA gene from 68 isolates from rhizosphere soil (black), non-rhizosphere soil (brown) and coralloid roots (green) of *Encephalartos villosus* grown in Oceanview and Rhebu in the Eastern Cape Province, South Africa.

### 3.3. Extracellular Enzyme Activities

There were insignificant differences in the N-cycling (nitrate reductase and N-acetylglucosaminidase) and P-cycling (acid and alkaline phosphatase) enzyme activities of rhizosphere and non-rhizosphere soils in Rhebu and Oceanview (Figure 4). Though the differences were insignificant, Rhebu and Oceanview rhizosphere soils had a higher alkaline phosphatase activity than non-rhizosphere soils (Figure 4C). In Rhebu, the non-rhizosphere soils had a higher acid phosphatase activity than the rhizosphere soils (Figure 4B).

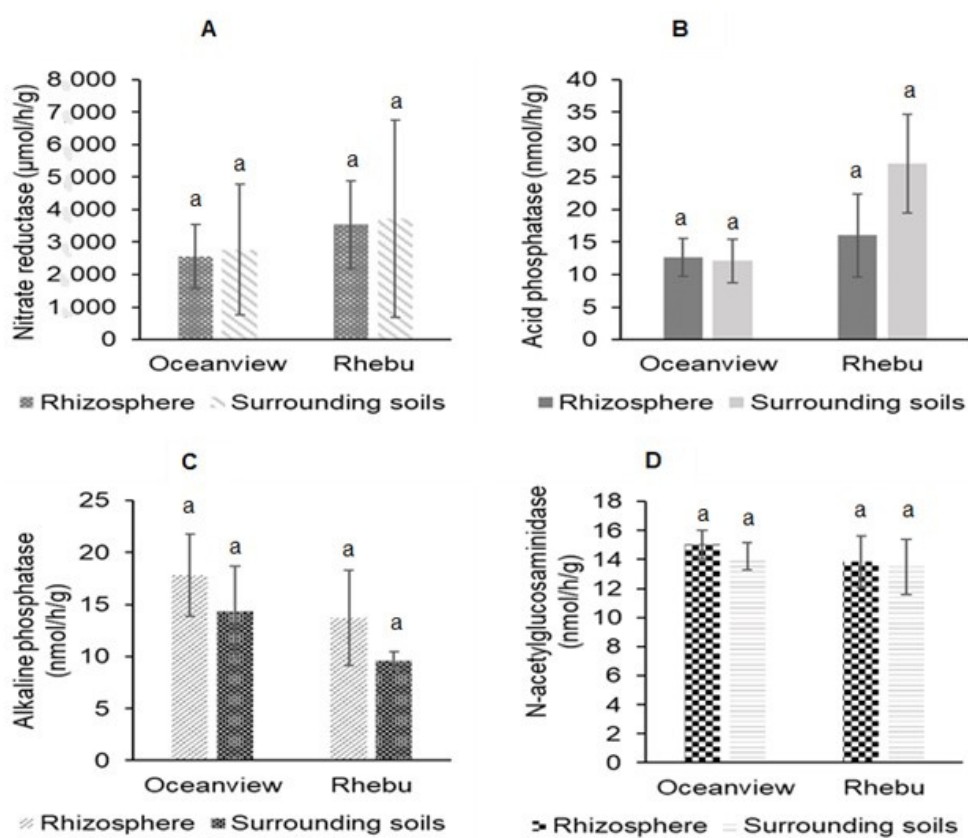

**Figure 4.** Soil extracellular enzyme activities in *Encephalartos villosus* rhizosphere and non-rhizosphere soils collected in Rhebu and Oceanview, Eastern Cape: (**A**) Nitrate reductase activity; (**B**) Acid phosphatase activity; (**C**) Alkaline phosphatase activity; (**D**) N-acetylglucosaminidase activity. In each graph, different letters denote significant differences in the enzyme activity of soil samples within study sites (independent samples *t*-test, $p \leq 0.05$, means ± SE, $n = 32$).

### 3.4. Leaf Nutrition and N Source Reliance

*Encephalartos villosus* leaf nutrition, number of leaves, leaf length, and N source reliance is represented in Figure 5. The mean leaf length of *E. villosus* plants in Rhebu and Oceanview was 186.5 ± 49.6 cm ($n = 8$) and 223.6 ± 28.5 cm ($n = 8$), respectively (Figure 5A). The sampled *E. villosus* plants from Rhebu and Oceanview had 11.4 ± 5.3 ($n = 8$) and 10.0 ± 4.2 ($n = 8$) leaves, respectively (Figure 5B). The Leaf N concentration of *E. villosus* plants growing in Rhebu and Oceanview was 2.29 and 2.13 mmol $g^{-1}$, respectively (Figure 5C). The P concentration of *E. villosus* leaves from Rhebu and Oceanview was 8.8 and 10.9 µmol $g^{-1}$, respectively (Figure 5D). The N concentration derived from the atmosphere by Rhebu plants was 0.60 and 0.70 mmol N $g^{-1}$ dw for Oceanview plants. Only 0.25 and 0.21 mmol N $g^{-1}$ dw was derived from the soil in Rhebu and Oceanview, respectively (Figure 5F). *Encephalartos villosus* plants growing in Rhebu and Oceanview utilised 70.31% and 77.00% N derived from the atmosphere, respectively (Figure 5E).

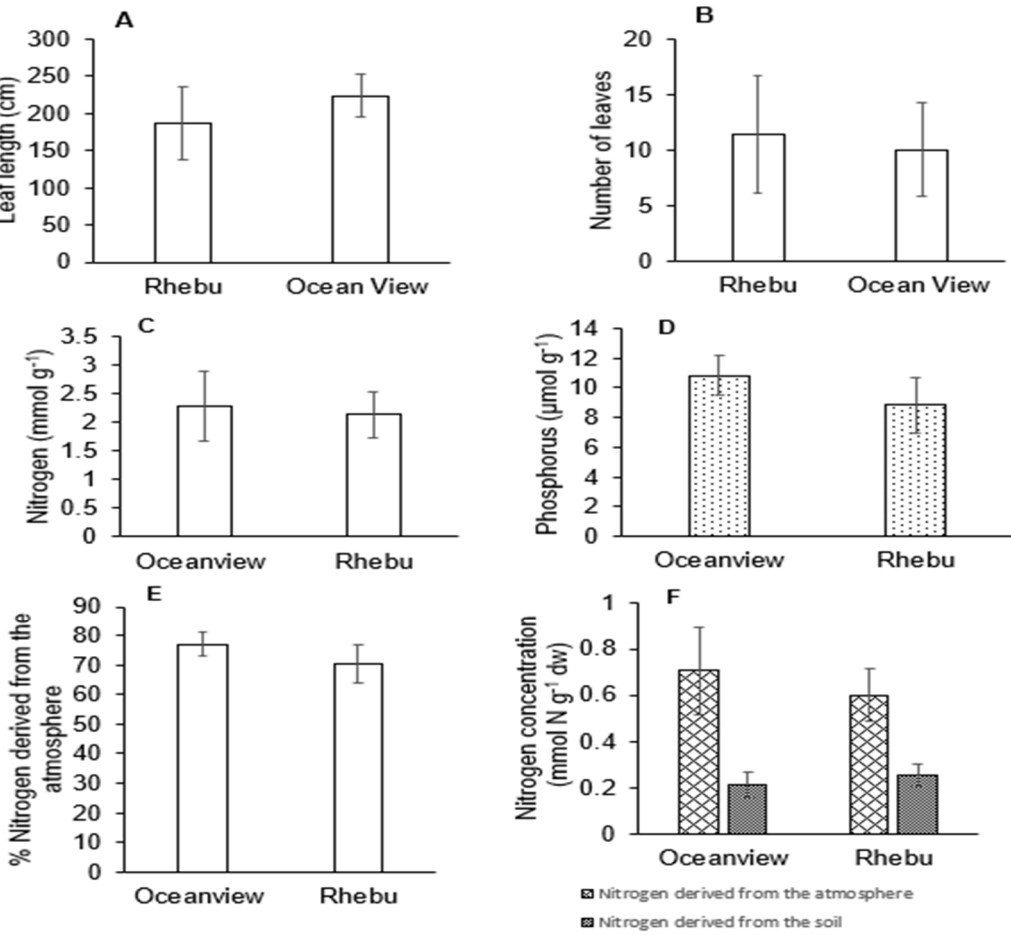

**Figure 5.** Leaf length (cm), number of leaves, Leaf nitrogen (mmol g$^{-1}$) and phosphorus concentration (μmol g$^{-1}$) and nitrogen source reliance of *Encephalartos villosus* plants growing in Rhebu and Oceanview, Eastern Cape. (**A**) Leaf length; (**B**) Number of leaves, (**C**) Leaf nitrogen; (**D**) Leaf phosphorus; (**E**) Percentage nitrogen derived from the atmosphere; (**F**) Nitrogen source reliance. The values represent the means ± SE, *n* = 16.

*3.5. Correlations between Extracellular Enzymes and Soil Nutrition in E. villosus Rhizosphere and Non-Rhizosphere Soils*

The correlation coefficients between extracellular enzymes and soil N and P in *E. villosus* rhizosphere and non-rhizosphere soils are represented in Table 1. Significantly positive correlations were observed between the soil P concentration and alkaline phosphatase activity in *E. villosus* rhizosphere and non-rhizosphere soils in Oceanview. Non-rhizosphere soils in Oceanview had a significantly positive correlation between soil P and acid phosphatase activity, and rhizosphere soils had insignificantly positive correlations between soil P and acid phosphatase activity. In Oceanview, significantly positive correlations were observed between soil N, nitrate reductase, and N-acetylglucosaminidase activity in rhizosphere and non-rhizosphere soils. In Rhebu, significantly positive correlations were observed between soil P, acid phosphatase, and alkaline phosphatase activity in rhizosphere and non-rhizosphere soils. The N-acetylglucosaminidase activity was significantly and positively correlated with soil N in rhizosphere and non-rhizosphere soils in Rhebu. The nitrate reductase activity of rhizosphere soils in Rhebu were significantly and positively correlated with soil N. Insignificantly positive correlations were observed between nitrate reductase activity and soil N in non-rhizosphere soils in Rhebu.

## 4. Discussion

In the present study, *E. villosus* coralloid roots and rhizosphere and non-rhizosphere soils were sampled to determine the role of cycad-microbe symbiosis, soil bacterial communities, and extracellular enzymes on the nutrition and growth of *E. villosus* in scarp forest ecosystems. The first noteworthy observation in this study is the clustering pattern of the isolated bacterial strains. Initially, one might hypothesize that bacterial clusters would align with their functional attributes. However, our findings demonstrate that these clusters are primarily dictated by the ecological niches from which the bacteria originated—specifically, rhizosphere soil, non-rhizosphere soil, and coralloid roots. We propose that the underlying reasons for this pattern are related to the concept of the filter theory [38,39] and its potential interaction with the hologenome theory [40]. The filter theory states that environmental filters, such as soil properties and abiotic factors, play a pivotal role in shaping microbial communities. Ref. [41] conducted an experiment on a corn-based annual cropping system and perennial switchgrass cropping system across three topographic positions and reported that soil properties associated with topographic position had a greater influence on the microbial composition than the plant species, showing how environmental filtering plays a role in determining microbial composition. Such an effect of environment on the design of the symbionts has been previously demonstrated in several legumes [42,43] and what is more, in soils where *E. villosus* grows [44].

In this study, it is worth highlighting that the soil properties and environmental conditions in the scarp forests, Oceanview and Rhebu, are very similar. These shared environmental traits would typically suggest that microbial communities in both locations should exhibit a higher degree of functional similarity due to comparable selection pressures. However, our findings show something different, implying that factors beyond these abiotic filters are at play. The hologenome theory [45,46] hypothesizes that the collective genetic material of a host organism and its associated symbionts, including bacteria, should be considered as a single functional unit or holobiont. As per the hologenome theory, the host's genotype, as well as its associated symbiotic microbial community, contribute to its ecological fitness and adaptation to specific niches. The holobiont contains mutualistic and beneficial microbes including plant growth promoting bacteria that enable the plant to survive in different environments [47]. The microbes in the holobiont play a pivotal role in the plant's response to abiotic and biotic stresses [48]. The plant host selects the composition of the beneficial microbial species by secreting organic compounds [49].

Therefore, variations in the holobiont's composition can drive the observed clustering patterns, even in environments with similar abiotic conditions. In the context of our study, it is likely that in a first step, the environment could have driven the kind of bacteria associated to *E. villosus*, as there are similarities in the bacterial strains identified in both Oceanview and Rhebu. Second, the host plants, in this case, the coralloid roots, play a pivotal role in shaping the microbial communities residing within their immediate vicinity. Soil microbial communities are influenced by plants through root secretions that provide microbes with energy, thus supporting rhizosphere colonisation [50,51]. Root secretions influence the selection of microbial communities in rhizosphere soils, resulting in rhizosphere microbial communities that are unique to the plant [52]. The uniquity of rhizosphere microbial communities and their respective plants aligns with the hologenome theory. Therefore, the selection pressure exerted by the secretions of *E. villosus* roots may have expanded beyond the coralloid roots and rhizosphere soils to non-rhizosphere soils. According to [44], the availability of nutrients in the soil may shape symbiotic interactions between plants and microorganisms, thus the abundance of P solubilising bacteria in *E. villosus* coralloid roots and rhizosphere and non-rhizosphere soils in both localities may have been driven by the low P concentration recorded in Table 1. Phosphorus solubilising bacteria belonging to the Pseudomonas, Bacillus, and Enterobacter genera have been reported to play a significant role in P solubilisation [53]. Phosphorus solubilising bacteria stimulate plant growth by producing phosphatases that mineralise and cycle P [54]. Ref. [55] studied the secretion of alkaline phosphatases and reported that *Bacillus licheniformis* secreted

30–40% of alkaline phosphatase in a medium containing low phosphate concentrations. Also, *Variovorax paradoxus*, a species isolated from Oceanview rhizosphere soils and Rhebu non-rhizosphere soils (Figures 1 and 2), reportedly produced 94.0 U/mL alkaline phosphatase in a low P liquid medium [56], elucidating that P solubilising bacteria secrete phosphatases in P deficiency.

According to the resource allocation model for extracellular enzyme activities, soil microbes exude extracellular enzymes to mineralise and cycle deficient soil nutrients [42]. The resource allocation model states that microorganisms allocate resources to synthesise enzymes involved in the acquisition of the most limiting element [46]; in the context of the current study, the limiting element is P. The resource allocation model was further verified by [57], who reported that the synthesis of phosphatases was dependent on P availability across various soils. Thus, the correlation between soil P and acid phosphatase activity (Table 2) may have been caused by P deficiency in the two scarp forest soils. The two scarp forests, Rhebu and Oceanview, sampled in this study were acidic (Table 1). In acidic soils, P forms insoluble complexes with aluminium (Al) and iron (Fe), making P unavailable for plant uptake [58]; this accounts for the low P concentrations recorded in the two scarp forests (Table 1). Ref. [11] reported positive correlations between soil extractable P, alkaline phosphatase, and acid phosphatase activity in the rhizosphere and non-rhizosphere soils of *Encephalartos natalensis* growing in P-deficient and acidic savanna woodland ecosystem soils, thus showing the link between soil nutrient deficiencies and extracellular enzyme activities. Additionally, positive correlations were recorded between N cycling enzymes (nitrate reductase and N-acetylglucosaminidase) and soil N; these findings are supported by [11], who reported correlations between N cycling enzymes and soil N in *E. natalensis* rhizosphere and non-rhizosphere soils. Since the resource allocation model states that microbes produce enzymes that will assist in the acquisition of the limiting element, we can deduce that the soils may have been N deficient as well. Nitrogen deficiency may have triggered BNF for N nutrition, which may account for the high reliance on NDFA by *E. villosus* in Oceanview and Rhebu. Thus, the reliance of *E. villosus* on N derived from the atmosphere may have been attributed to N deficiency and P provision through P-solubilising bacteria, extracellular enzyme activities, and possibly arbuscular mycorrhizal fungi.

**Table 2.** Pearson's correlation coefficient values between chosen soil parameters and soil enzyme activities.

| | Oceanview | | | | Rhebu | | | |
|---|---|---|---|---|---|---|---|---|
| | Rhizosphere Soils | | Non-Rhizosphere Soils | | Rhizosphere Soils | | Non-Rhizosphere Soils | |
| | N | P | N | P | N | P | N | P |
| Nitrate reductase (μmol/h/g) | 0.89 * | - | 0.98 * | - | 0.94 * | - | 0.70 | - |
| N-acetylglucosaminidase (nmol/h/g) | 0.93 * | - | 0.98 * | - | 0.78 * | - | 0.89 * | - |
| Acid phosphatase (nmol/h/g) | - | 0.68 | - | 0.96 * | - | 0.98 * | - | 0.84 * |
| Alkaline phosphatase (nmol/h/g) | - | 0.98 * | - | 0.97 * | - | 0.97 * | - | 0.76 * |

Significant values are represented by *.

Cycads have been extensively reported to associate with cyanobacteria such as *Nostoc*, *Scytonema*, and *Richelia* species, which fix atmospheric N for plant uptake [59,60]. Other studies have reported the presence of non-cyanobacterial species belonging to the *Rhizobium*, *Mesorhizobium, Bradyrhizobium,* and *Burkholderia* genera [20,25]. Though cyanobacterial species were not isolated from the coralloid roots, the presence of non-cyanobacterial species may have contributed to the reliance of *E. villosus* on NDFA. Nitrogen fixing bacteria belonging to the *Lysinibacillus, Paenibacillus, Stenotrophomonas, Rhizobium*, and

*Enterobacter* genera isolated from the coralloid roots, have been reported to be associated with cycad species *Dioon edule* [25] and *E. natalensis* [11]. Ref. [20] reported that cycads are predominantly associated with rhizobial species however, in this study, *E. villosus* was associated with rhizobial and non-rhizobial bacteria. According to [61], the coexistence of rhizobial and non-rhizobial bacteria improves N fixation and nodulation in legume hosts, which may be true for cycads as well. Ref. [62] reported that non-rhizobial bacteria enhance atmospheric N fixation because of their increased efficiency in N fixing which may have led to the high reliance of *E. villosus* on NDFA.

## 5. Conclusions

Our findings challenge the conventional expectation that environmental filters alone dictate microbial community composition in similar habitats. Instead, they suggest that host-microbe interactions, as proposed by the hologenome theory, are significant drivers of microbial community structuring. This study shows how the wide distribution of *E. villosus* in nutrient deficient and acidic ecosystems may be linked to the cycad's association with bacteria with plant growth-promoting traits. Furthermore, this study revealed that *E. villosus* rhizosphere and non-rhizosphere soil bacterial communities and their associated enzyme activities contribute to soil nutrition in nutrient deficient and acidic scarp forests. To our knowledge, this study is the first to determine the N source reliance in cycads. These findings open a new avenue of study focused on cycad-microbe symbiosis, which will provide a result-based analysis of N-fixation in cycads. If the contributions of non-cyanobacterial cycad symbionts in N-fixation efficiency are determined, it would indicate that the N source reliance of cycad species is not only limited to cyanobacteria but also non-cyanobacterial species.

**Author Contributions:** Conceptualization, A.M. and T.N.S.; methodology, N.M., T.N.S., M.A.P.-F. and A.M.; validation, N.M., T.N.S., M.A.P.-F. and A.M.; formal analysis, N.M. and M.A.P.-F.; writing—original draft preparation, N.M., T.N.S., M.A.P.-F. and A.M.; writing—review and editing, N.M., T.N.S., M.A.P.-F. and A.M.; supervision, T.N.S. and A.M.; project administration, N.M. and A.M.; funding acquisition, A.M. and T.N.S. All authors have read and agreed to the published version of the manuscript.

**Funding:** The research leading to these results received funding from the National Research Foundation under the grant agreement number (Grant UID 129403 and 138091).

**Data Availability Statement:** The data and material can be made available upon request.

**Acknowledgments:** We appreciate the financial support from the National Research Foundation (Grant UID 129403 and 138091). We acknowledge the support of the University of KwaZulu-Natal (School of Life Sciences) and University of Mpumalanga (School of Biology and Environmental Sciences).

**Conflicts of Interest:** We declare no known competing financial and non-financial interests with regards to the current research. The funders had no role in the design of the study; in the collection, analyses, or interpretation of data; in the writing of the manuscript, or in the decision to publish the results.

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
