# Peer review of "Exploring the Influence of Ecological Niches and Hologenome Dynamics on the Growth of Encephalartos villosus in Scarp Forests"

_soilsystems, doi:10.3390/soilsystems8010021_

Round 1

Reviewer 1 Report

Comments and Suggestions for Authors

The title of the article is confusing, could it be nutrient deficient soils? But deficient in all nutrients?

The title of the work is not in line with the objective of the work.

The objective of the work was to verify the association of bacteria in the production of Encephalartos villosus

In nutrient-poor acidic soils. At no point do the authors report the association with bacteria in the title.

Introduction

The introduction is adequate.

M&M

The description of the soil analysis methodology is confusing. P and K extracts cannot be read in Atomic Absorption Spectrometry, before extracting these elements in a chemical solvent or using a resin. What would Ambic-2 be? resin or solvent?

The analysis methodology in South Africa is very different from the rest of the world, in articles I have never seen the stirring of solutions in Stirrer, for P, K, Ca and Mg, described. How can the authors justify this? because there is no way to compare the results if the analytical methods are very different.

Furthermore, the authors did not classify the soil according to international standards such as FAO's Soil taxomony

Results

The authors say that the soils are poor in nutrients, but what did they compare it to? The only low nutrient is P.

Tables missing units

Figures need to improve quality.

Dicussion

The bacteria characterization part is adequate

However, the chemical characteristics of the soil were little discussed. The authors do not discuss why these soils are acidic or poor in nutrients. The authors already directly report the issue of data on soil microorganisms and do little to relate them to nutrient content in the soil.

For me it is clear that the objective of the work was not the chemical characteristics of the soil but biological ones.

Conclusion

The authors cannot maintain that the soils are poor and acidic

The conclusion is very extensive and relates more to the issue of future research, which is unnecessary.

Author Response

Comments and Suggestions for Authors

The title of the article is confusing, could it be nutrient deficient soils? But deficient in all nutrients?

Author response- Title was modified and changed. Nutrient deficiency noted was only phosphorus deficiency as it was the lowest nutrient concentration recorded in both scarp forests.

The title of the work is not in line with the objective of the work.

Author response- Title was changed and aligned to the objectives of the study.

The objective of the work was to verify the association of bacteria in the production of Encephalartos villosus.

Author response: Objectives modified.

In nutrient-poor acidic soils. At no point do the authors report the association with bacteria in the title.

Author response: Title was changed to accommodate the association and clustering pattern of the isolated bacteria.

Introduction

The introduction is adequate.

Author response- Introduction proofread only.

M&M

The description of the soil analysis methodology is confusing. P and K extracts cannot be read in Atomic Absorption Spectrometry, before extracting these elements in a chemical solvent or using a resin. What would Ambic-2 be? resin or solvent?

Author response: Ambic-2 was used to extract potassium, phosphorus, zinc, copper, and manganese. In this context, ambic-2 is a solvent. Methodology was rewritten to address the confusion and more details on the extraction and determination of the nutrient concentrations.

The analysis methodology in South Africa is very different from the rest of the world, in articles I have never seen the stirring of solutions in Stirrer, for P, K, Ca and Mg, described. How can the authors justify this? because there is no way to compare the results if the analytical methods are very different.

Author response: The methodology was written as per the KwaZulu Natal Department of Agriculture and Rural Development Analytical services for soil nutrient concentrations, total cation concentrations, exchange acidity, and pH analysis using the methodology using protocols explained by Manson and Roberts (2000). The methodology includes the stirring or solutions using a stirrer. The methodology used for the soil characteristics was the same for both Rhebu and Oceanview soils.

Furthermore, the authors did not classify the soil according to international standards such as FAO's Soil taxonomy

Author response: Within the context of our study, which focused on soil microbes and nutrients, it was impossible to classify the soil types of the scarp forests. This is because we sampled only the top soil horizon (0 – 20cm) where microbial activity takes place. The sub soil horizons that are free from microbial activities were not sampled, and we were therefore not able to classify the soil following the FAO soil taxonomy or WRB. Also the study did was not about pedology, making it impossible for us to classify the soil.

Results

The authors say that the soils are poor in nutrients, but what did they compare it to? The only low nutrient is P.

Author response: Phosphorus was the only nutrient regarded as deficient.

Tables missing units

Author response: Units added in table 1.

Figures need to improve quality.

Author response: Figures reinserted in the text and quality checked.

Discussion

The bacteria characterization part is adequate

Author response: No changes made.

However, the chemical characteristics of the soil were little discussed. The authors do not discuss why these soils are acidic or poor in nutrients. The authors already directly report the issue of data on soil microorganisms and do little to relate them to nutrient content in the soil.

Author response- The soils are said to be acidic because when they were analysed their pH was below 7, qualifying the soils as acidic. According to Mucina et al. (2018), scarp forests are characterised as nutrient deficient. The forests sampled in this study were from scarp forests in the Eastern Cape and deduced to be nutrient poor as per remarks made by Mucina et al. (2018). Also, the very low phosphorus concentration showed deficiencies in the soils.

For me it is clear that the objective of the work was not the chemical characteristics of the soil but biological ones.

Author response- The chemical characteristics of the soils were linked to soil extracellular enzymes and this connection was proven by positive and significant correlations as illustrated in Table 2.

Conclusion

The authors cannot maintain that the soils are poor and acidic.

Author response- Soil samples were collected in scarp forests which are characterised as nutrient deficient. Furthermore, the phosphorus concentration was much lower than the other nutrients assayed, and the pH recorded was less than 7 qualifying the soils as acidic.

The conclusion is very extensive and relates more to the issue of future research, which is unnecessary.

Author response- Information on future research was removed from the conclusion.

Reviewer 2 Report

Comments and Suggestions for Authors

RE: Untangling the distribution of Encephalartos villosus in nutrient deficient soils of South Africa

 The manuscript provides valuable insights into the intricate relationship between host-microbe interactions, microbial community composition, and nutrient acquisition in Encephalartos villosus, a cycad species, in acidic and nutrient-deficient scarp forest ecosystems. The paper is well-structured and presents a comprehensive analysis of the research findings. However, the proposed mechanisms driving microbial community assembly and their impact on nutrient cycling are intriguing and warrant further exploration, thus incorporating the following comments, will improve the clarity of the manuscript

1.       Clustering Pattern of Bacterial Strains: The observation that the clustering pattern of isolated bacterial strains is primarily dictated by the ecological niches from which they originated (rhizosphere soil, non-rhizosphere soil, and coralloid roots) is a significant finding. The authors appropriately propose that host-microbe interactions may be a key driver of this pattern, in line with the hologenome theory. This hypothesis is intriguing and provides a novel perspective on the assembly of microbial communities. However, it would be helpful if the authors could expand on how this concept aligns with or differs from existing theories on microbial community assembly. Additionally, further exploration into the specific mechanisms driving these host-microbe interactions would enhance the paper's depth.

2.       Environmental Filters and Host Genetics: The discussion of how both environmental filters and host genetics influence microbial community composition is insightful. However, it would be beneficial to provide more context and examples to help readers understand the interplay between these factors in shaping the observed patterns. Additionally, mentioning any relevant studies or systems where similar interactions have been observed would strengthen the paper's arguments. This section should be improved by providing the relevant information and mechanisms for the claim the author had made.

3.       Nutrient Availability and Extracellular Enzyme Activities: The paper effectively links nutrient availability in soil to the presence of specific bacteria, such as phosphorus (P)-solubilizing bacteria. This connection is well-supported by the correlation between soil P and acid phosphatase activity. While the paper highlights P deficiency, it would be valuable to discuss how this links to the broader context of nutrient cycling and potential impacts on plant growth. A more detailed explanation of the resource allocation model for extracellular enzyme activities would also aid readers in understanding the importance of these findings.

4.       N Source Reliance in Cycads: The revelation that cycads may rely on non-cyanobacterial species for nitrogen (N) fixation is a noteworthy contribution. However, It would be beneficial to discuss how this reliance on non-cyanobacterial species could affect N availability, cycling in these ecosystems, and any broader ecological consequences.

5.       Use uniform terminology, some time you have written non-rhizosphere soil, other time surrounding soil, better to change to non-rhizosphere soil

6.       Fig 5 is not clear, what do you mean by N concentration, also irrelevant parameters have been group together in a single Fig.

7.       Fig 6, an explanation of total 31.6+16= 48% is far lower than what we need for such type of studies, I suggest to delete this and its information, rather its correlation coefficient may be provided in a tabular form

8.       No of references should be kept around 40, the current number is more than required for a journal article.

9.       The formula and procedure for calculating the % NDFA needs to be re-checked once again, and also the units should be checked as per mill or percent.

Comments on the Quality of English Language

Fine 

Author Response

Reviewer 2

Comments and Suggestions for Authors

RE: Untangling the distribution of Encephalartos villosus in nutrient deficient soils of South Africa

 The manuscript provides valuable insights into the intricate relationship between host-microbe interactions, microbial community composition, and nutrient acquisition in Encephalartos villosus, a cycad species, in acidic and nutrient-deficient scarp forest ecosystems. The paper is well-structured and presents a comprehensive analysis of the research findings. However, the proposed mechanisms driving microbial community assembly and their impact on nutrient cycling are intriguing and warrant further exploration, thus incorporating the following comments, will improve the clarity of the manuscript.

  1. Clustering Pattern of Bacterial Strains: The observation that the clustering pattern of isolated bacterial strains is primarily dictated by the ecological niches from which they originated (rhizosphere soil, non-rhizosphere soil, and coralloid roots) is a significant finding. The authors appropriately propose that host-microbe interactions may be a key driver of this pattern, in line with the hologenome theory. This hypothesis is intriguing and provides a novel perspective on the assembly of microbial communities. However, it would be helpful if the authors could expand on how this concept aligns with or differs from existing theories on microbial community assembly. Additionally, further exploration into the specific mechanisms driving these host-microbe interactions would enhance the paper's depth.

Author response- The hologenome theory was aligned with the rhizosphere colonization theory which posits that soil microbial communities are influenced by plants through root secretions that provide microbes with energy, resulting in rhizosphere soils that are unique to the plant they are associated to.

  1. Environmental Filters and Host Genetics: The discussion of how both environmental filters and host genetics influence microbial community composition is insightful. However, it would be beneficial to provide more context and examples to help readers understand the interplay between these factors in shaping the observed patterns. Additionally, mentioning any relevant studies or systems where similar interactions have been observed would strengthen the paper's arguments. This section should be improved by providing the relevant information and mechanisms for the claim the author had made.

Author response- More information on the filter theory and host genetic was added to provide more context. An example of the filter theory was added.in the discussion.

  1. Nutrient Availability and Extracellular Enzyme Activities: The paper effectively links nutrient availability in soil to the presence of specific bacteria, such as phosphorus (P)-solubilizing bacteria. This connection is well-supported by the correlation between soil P and acid phosphatase activity. While the paper highlights P deficiency, it would be valuable to discuss how this links to the broader context of nutrient cycling and potential impacts on plant growth. A more detailed explanation of the resource allocation model for extracellular enzyme activities would also aid readers in understanding the importance of these findings.

Author response- A more detailed explanation of the resource allocation model for extracellular enzyme activities was added in the discussion.

  1. N Source Reliance in Cycads: The revelation that cycads may rely on non-cyanobacterial species for nitrogen (N) fixation is a noteworthy contribution. However, It would be beneficial to discuss how this reliance on non-cyanobacterial species could affect N availability, cycling in these ecosystems, and any broader ecological consequences.
  2. Use uniform terminology, some time you have written non-rhizosphere soil, other time surrounding soil, better to change to non-rhizosphere soil 

Author response- Surrounding soils changed to non-rhizosphere soils.

  1. Fig 5 is not clear, what do you mean by N concentration, also irrelevant parameters have been group together in a single Fig.

Author response- N concentration is Leaf nitrogen (N) concentration. As it is stated on the graph itself, N concentration refers to the milli molar concentration of N per gram of dry mass (i.e., the total concentration of N in the sampled leaves). Parameters were grouped together as growth characteristics (leaf length and number of leaves) and leaf nutrient concentrations (nitrogen, phosphorus, % nitrogen derived from the atmosphere (%NDFA), nitrogen source reliance (NDFA and NDFS)) that were analysed in E. villosus sampled leaves.

  1. Fig 6, an explanation of total 31.6+16= 48% is far lower than what we need for such type of studies, I suggest to delete this and its information, rather its correlation coefficient may be provided in a tabular form

Author response- PCA removed and replaced with simple linear correlation analysis represented in tabular form.

  1. No of references should be kept around 40, the current number is more than required for a journal article.

Author response- Due to the extensive research done on cycads and cycad-microbe symbiosis more information was available. That coupled with the references from the methodology and discussion led to the reference section having more than 40 references. Though the number of references was reduced in the introduction, the new information added in the discussion led to more references being added.

  1. The formula and procedure for calculating the % NDFA needs to be re-checked once again, and also the units should be checked as per mill or percent.

Author response- The formular and procedure were double checked, and the unit mentioned.

Reviewer 3 Report

Comments and Suggestions for Authors

The work concerns untangling the distribution of Encephalartos villosus in nutrient deficient soils of South Africa. I think that the research topic is interesting and it is good that the authors conducted such an experiment. However, the work is not very interesting.

My comments:

1. Authors must emphasize the purpose of the research and describe what is new in their research.

2. Material and methods. Some of the research was performed by other units. Please describe the methods in detail and indicate the names of the equipment.

3. The figures show the results of statistical analyzes - the letters indicate different homogeneous groups. In Figure 4 B, 4C there are large differences in the values and the same group is shown (marked with the letter a). Was the statistical analysis performed correctly?

4. The discussion is very short. There are not many references to other works. The word "suggests" appears many times in the discussion.

5. The summary does not contain any relevant information. It is very general.

6. There are editorial errors in the work - they are marked in yellow.

Author Response

The work concerns untangling the distribution of Encephalartos villosus in nutrient deficient soils of South Africa. I think that the research topic is interesting and it is good that the authors conducted such an experiment. However, the work is not very interesting.

Author response- Thank you for the comment, however, we think conducting a study on the role of cycad-associated microbes on E. villosus growth and nutrition is essential because E. villosus has a life history similar to cycads that cannot be studied due to their conservation status. Therefore, studying the role of microbes on E. villosus growth will not only shed light on how microbial interactions in E. villosus contribute to soil nutrient inputs, growth, and plant nutrition but will also give insights into other cycad species with similar life history. Also, these kinds of manuscripts are very interesting from the ecological point of view as they provide evidence on the behavior of native species in nature. The study has implication for plant conservation, what is relevant in the current situation of global change.

My comments:

  1. Authors must emphasize the purpose of the research and describe what is new in their research.

Author response- The objective/purpose of the research is clearly written in the last paragraph of the introduction.

  1. Material and methods. Some of the research was performed by other units. Please describe the methods in detail and indicate the names of the equipment.

Author response- Most of the analysis was outsourced as per the materials and methods section, thus details pertaining to the exact equipment used during the analyses are missing. However, in instances where the analysis was carried out by us the equipment used was listed (e.g., enzyme activities). In instances where a detailed methodology was made available, the methodology and information on the equipment used was made available in the manuscript (Please see page 5 lines 234-248)

  1. The figures show the results of statistical analyzes - the letters indicate different homogeneous groups. In Figure 4 B, 4C there are large differences in the values and the same group is shown (marked with the letter a). Was the statistical analysis performed correctly?

Author response- The statistical analysis was redone twice and still there were no significant differences in the enzyme activities of rhizosphere and non-rhizosphere soils.

  1. The discussion is very short. There are not many references to other works. The word "suggests" appears many times in the discussion.

Author response- Number of times the word “suggests” is used was reduced and replaces with terms such as “states”. Discussion length increased and other works referenced.

  1. The summary does not contain any relevant information. It is very general.

Author response- Information on the clustering pattern of the isolated bacteria and its alignment with the hologenome theory noted in the abstract.

  1. There are editorial errors in the work - they are marked in yellow.

Author response- All editorial errors were noted and corrected.

Round 2

Reviewer 1 Report

Comments and Suggestions for Authors

The authors made corrections to the article.

The article presents important information for plant nutrition.

The article is suitable for publication.

Reviewer 2 Report

Comments and Suggestions for Authors

The author have incorporated most of my comments, and therefore I am accepting the manuscript in its present form 

Reviewer 3 Report

Comments and Suggestions for Authors

My comments have been taken into account.